# Cycle of Perpetual Vulnerability for Women Facing Homelessness near an Urban Library in a Major U.S. Metropolitan Area

**DOI:** 10.3390/ijerph17165985

**Published:** 2020-08-18

**Authors:** Janny S. Li, Lianne A. Urada

**Affiliations:** 1School of Social Work, San Diego State University (SDSU), San Diego, CA 92182, USA; janny.siuling.li@gmail.com; 2Department of Medicine, University of California San Diego (UCSD), La Jolla, CA 92093, USA

**Keywords:** women, homelessness, sexual exploitation, human trafficking, violence, vulnerability

## Abstract

Background: Homelessness among women and the multiple vulnerabilities they endure (sexual exploitation/human trafficking, violence, and mental health issues) is a perpetually unresolved issue in the U.S. and globally. Methods: This study is based on qualitative in-depth interviews accompanied by brief socio-demographic surveys conducted among 32 total participants, consisting of cisgender females (*n* = 17) and cisgender males (*n* = 15) experiencing homelessness at a large public library. Results: Of the women, 35% were White, 35% Latina, 18% African American/Black, and 18% LGBT. Half of all participants said in qualitative interviews that they witnessed violence against women, and/or experienced unwanted harassment/sexual exploitation; one in three described suspected human trafficking. Of the women interviewed, half struggled with mental health symptoms, feelings of hopelessness, and nearly all reported isolation; approximately one-third had substance use issues. Many described an inadequate number of emergency and long-term shelters Available for women facing homelessness; many had to wait or saw other women waiting to get into shelters and faced abuse on the streets in the meantime. Conclusion: The emergent themes showed that women face a “cycle of perpetual vulnerability” with three relational pathways: iterated trauma from chronic abuse/violence inflicted on them, a state of paralysis due to inadequate availability of supportive services, shelters, and mental health resources to cover all women living on the streets, leaving women susceptible to being a target phenotype for predators.

## 1. Introduction

In the 2019 Annual Assessment Homeless Report (AAHR) to Congress, the US Department of Housing and Urban Development completed a point-in-time study where 8102 individuals were captured as experiencing homelessness in a large southern California city, ranking the city among the top five in terms of homeless residents in the nation [1]. The term homelessness is defined as “lacking a fixed, regular, and adequate nighttime residence” [1]. This includes individuals living on the streets, in vehicles, shelters, temporary housing, couch surfing, and with friends. While a lack of stable residency is at the root of homelessness, the issue has evolved into a public health crisis, one with compounding conditions that require a multitude of solutions beyond housing. People experiencing homelessness suffer from higher levels of personal disability, including chronic health conditions and mental health issues. Thus, their time living on the streets can further propel them into profound estrangement from their support system and society [2], making them an extremely vulnerable population.

Vulnerability can be broadly understood as susceptibility to danger, harm, or death. Social scientists define “vulnerable groups” as those that experience a “greater-than-average risk of developing health problems by virtue of their marginalized sociocultural status, their limited access to economic resources, or their personal characteristics such as age and gender” [3]. It’s application in policies and procedures can be found in tools such as the Vulnerability Index, which uses markers to identify and prioritize housing for homeless individuals who are at a higher risk for mortality [4]. Markers that inform their vulnerability assessment include number of hospitalizations, age, substance use, psychiatric diagnoses, and other health factors.

Gendered vulnerability is an important theme in the issue of homelessness. Given the historical discrimination, marginalization, sexual harassment, and disempowerment of women, homeless women continue to face added vulnerabilities compared to homeless men. Women make up 25% of the homeless population in the southern California city in which this study took place, though it is steadily on the rise [5]. Between 2018 and 2019, the US’s homeless female population jumped 8%, representing 39% of the total homeless population [1]. Experts suspect women’s numbers are still likely under reported because women and girls may avoid the streets and shelters where community outreach or the homelessness count takes place. The streets, alleyways, and outskirts of public places where homeless individuals frequent are a male-dominated sphere, making women a minority group and vulnerable to violent victimization or sexual assault [6]. Similarly, women report feeling unsafe in shelters, having experienced abuse or been offered substances in these co-ed facilities [7]. In the last year, unsheltered women and girls grew by 12% [1]. Instead of shelters, women might seek refuge with friends, family, couch surf, or live in their vehicles [7], making them less visible in the public view of homelessness. Their needs can be overlooked and underestimated. Women who are homeless face major vulnerabilities, including violent victimization, sexual assault, and trauma.

In the District of Columbia, a point-in-time study reported that 54% of homeless women had experienced violent victimization, 72% had mental health issues, 28% also experienced sexual assault, and 36% had been trafficked [8,9]. In one study interviewing 993 homeless young people across 13 cities in the United States and Canada, approximately 40% were sexually exploited young women [10]. A homeless woman’s risk for sexual assault is 20% greater than women of the general public [2]. In a systematic review of 25 studies evaluating homelessness, sexual assault, and outcomes, several reported on negative mental health outcomes associated with physical and sexual assault, including PTSD, severe psychiatric symptoms, anxiety and mood disorders [11]. Homeless women who experienced a history of abuse had higher rates of anxiety disorders compared to those who did not experience abuse—54.8% to 15.4%, respectively [12]. Compared to their male counterparts, homeless women are 1.5 times at risk of suicide ideation [13]. Another study found 74% of women in domestic violence shelters needed mental health services [14]. Similarly, mothers represent a significant subpopulation of homelessness with distinct needs, prospects, and safety concerns. Approximately 50% of women at domestic violence shelters were mothers traveling with children [14]. Among women who had been homeless for 2 years and longer, women with children experienced higher levels of depression and substance dependency compared to homeless women without children [15].

This article further examines the lesser known social and structural reasons for vulnerabilities for homelessness among women such as the lack of Available women-specific shelters and supportive services. The aims of this study were to (1) investigate vulnerabilities for women, including violence, sexual assault, and human trafficking risks; (2) identify existing barriers for women, including shelters, childcare, and domestic violence resources; (3) explore the relationship between mental health and homelessness for women.

Furthermore, a critical gap exists in theoretical work that encompasses the relational sequelae of homelessness, or an interdependent pattern that keeps homeless women in perpetual vulnerability. A similar model can be found in the intimate partner violence theory known as *Cycle of Violence*, which describes a repetition of abuse and emotional response [16], yet few studies until now have evaluated a cyclical pattern of conditions for women facing homelessness that may keep them in a disempowered state.

## 2. Methods

### 2.1. Study Setting

Public libraries have become a frequented space for those struggling with homelessness. Libraries present fewer restrictions for those in shelters or for those who are unsheltered and have nowhere to stay during the day. Those experiencing homelessness are also attracted to the library’s free access to computers, books, and the internet. Overall, the space is a quiet, clean, and safe escape from the bustle of the city and police interactions. The library, located in a major metropolitan area on the U.S. West Coast, became a research partner to recruit study participants (January–June 2019). For recruitment, a signup sheet was Available at the 3rd floor library reference desk for patrons interested in participating in the study. Participants were recruited through flyers at the library, by library staff announcements over the intercom, or by researchers at the entrance upon opening hour.

### 2.2. Study Sample and Data Collection Procedures

This study used qualitative in-depth interviews of women (*n* = 17) and men (*n* = 15) struggling with unstable housing and socio-demographic data (age, ethnicity, gender identity, sexual orientation, number of dependents/children) collected from a brief, one-page survey collected at the time of the interview (Table 1). An interview guide for in-depth interviews included 10 major questions/domains with probe/follow-up questions on the barriers to stable housing, feelings of isolation, human trafficking risks, and recommendations for policy changes (Appendix A). In total, 32 library patron participants were recruited through fliers distributed at a large urban public library. This study utilized interviews from both male and female respondents to complete a thematic analysis pertaining to women. While men struggling with homelessness may also encounter risks such as violence or vulnerabilities, those aims are beyond the scope of this paper. This study focused on female vulnerabilities and risks. Female respondents provided their observations and direct experiences, while the observations of male respondents helped supplement the female narrative and contextualize the experiences of women. Thus, male testimonials were analyzed and were only included in their observations of female vulnerabilities and risks. The study protocol was approved by the San Diego State University (SDSU) Institutional Review Board.

Participants in the study received Metropolitan Transportation Authority Day Passes, valued at $20 total for their participation. Seven interviewers conducted the individual in-depth interviews with the library patrons. Four were Master’s level graduate students (three in Social Work, one in Public Administration/Latin American Studies), two were doctorate students specializing in Substance Use, and one was the lead Principal Investigator (professor).

Interviewers scheduled a private study room at the library to conduct the interviews, or to complete interviews in the patio or roof with participants. Interviews were audio-recorded with informed consent from participants. The interviews were then transcribed without identifying information. Interviews lasted between 1 and 2 h with participants, during which interviewers also took brief notes. Upon completion, the Principal Investigator of the study reviewed the transcriptions within 7 days to provide feedback to interviewers to ensure study protocol and quality, and to review data for any potential codes. For non-English speakers, interviewers asked or translated the questions, then interviewers transcribed the answer on the survey and in the interview guide.

### 2.3. Data Analysis

This study used a grounded theory [17,18] and thematic analysis to generate new codes for major recurring themes that emerged from the qualitative interview data. Inter-coder reliability across coding was reached via a standard approach [19]. Two researchers worked independently to code the transcripts and then met to agree on codes that did not fit with the predetermined deductive themes. The researchers then identified new inductively derived themes not yet well described in the literature. Our framework, the Cycle of Perpetual Vulnerability Theory, described at the end of the Section 3 (Figure 1) arose out of these domains. The themes were grouped into three relational pathways: *target phenotype, iterated trauma,* and *state of paralysis*.

## 3. Results

Half of the participants were White with the rest Black/African American (*n* = 7), Hispanic/Latino (*n* = 7), and other (*n* = 6) (Table 1). According to the brief surveys, 47% of the women resided in shelters and 41% lived on the streets. From the in-depth interviews, 13 were mothers, three were traveling with children, and six were seeking domestic violence shelters.

Our findings from the qualitative interviews found two major themes: (1) violence against women, categorized as sexual exploitation, violent victimization, intimate partner violence; (2) social and structural barriers, categorized as limitations in shelters, childcare services, and mental health resources. Social and structural barriers were defined as systemic gaps in agencies or community support that prevented homeless individuals from accessing services or needs. We reported on the following themes based on our female in-depth interviews. Male testimonials were only included in the subcategories of sexual exploitation and violent victimization under the major theme, violence against women.

### 3.1. Violence against Women

Through qualitative interviews, both male and female participants shared accounts of sexual exploitation, violent acts, and intimate partner violence witnessed toward women. Half of all male and female participants (*n* = 16) witnessed or experienced violent acts against women. Both male and female interviewees report that offenders look for specific characteristics in women, including being “hooked” on drugs, requiring financial assistance, being alone, among others. Both male and female participants stated that women facing homelessness are disproportionately approached for sexual acts. Nearly half of our female participants (*n* = 7) experienced intimate partner violence. We report on the following themes: sexual exploitation, violent victimization, and intimate partner violence.

#### 3.1.1. Sexual Exploitation/Human Trafficking

Women residing on the streets reported sexual assault, verbal harassment, and unwanted physical contact compared to the women living in shelters. A female respondent shared accounts of being harassed in her sleep. (*“…I get people that will come up and like be touching on me while I’m asleep and I’ll wake up to that…”*) A female respondent confirmed that she had been forced and deceived into exchanging sex for money (otherwise defined as human trafficking), stating that offenders look for certain characteristics, such as a woman using substances. (*“…Most of the time, the predators will try to find somebody who’s already hooked…on any kind of drug…”*) Another female respondent stated that predators and offenders lure women through prospects of a job or safe shelter. (*“… I have watched it, selling it as having jobs or cheap rooms for rent. I think we should make others aware of who they are so they can stop targeting these women…”*)

Regarding suspected human trafficking, the same respondent reported:
“Outside the [shelter] at the bus stop, there’s this dude that always shows up…and he’s telling the girls, ‘Oh, you can get a job working with me cleaning a trailer,’ and this and that, ‘I’m looking for a house cleaner.’ And [my friend] told me that two girls went with him and that she never saw them again…When [my friend] was talking to me…he went up to her…was just making general threats. And I backed off away from him, and I knew, I was like, ‘Uh-huh, he’s looking for people to take,’ and he’s targeting people—the women from there that are homeless with the [shelter]. Does [the shelter] know?”

In total, *11 men and women reported sex trafficking risks against women*. One male respondent stated that the stakes for homeless women are higher. (*“...I think they have it so much harder than men. You could be raped, there are so many worse things that could happen. And I think their fall is a lot quicker...”*)

Men observed that women were disproportionately approached or kidnapped for sexual acts. (*“…A man can pretty much blend in with other men but a woman, especially a younger woman, she’s a target...”*) Another male interviewee stated:
“I have seen people get kidnapped. Mainly women. They just take them and you don’t see them again, I mean these women are lost in drugs and nobody looks for them after that.”

One woman shared a personal experience of being sexually assaulted while living on the streets. She had been deceived by a friend, who had given her an adulterated drink. Shortly after, she lost consciousness.
“I woke up under the bridge over that way towards (uh) you could see the Coronado bridge… and there was this big old…guy next to me and I had bruises up my arms and it looked like needle marks…. I don’t use needles…I woke up like nude from the waist down.”

The puncture marks from the needle use swelled and became infected the next day. She woke up “terrified” and underwent “shakes.” She did not report the crime to authorities. (*“…if you say anything to the police…you get what’s called a snitch jacket. Once you get that, you’re automatically greenlighted…”*)

#### 3.1.2. Violence Victimization—“Did Anyone Ever Threaten You/Them with Violence or Other Ways?”

Violence against women was a prevalent theme among our male and female respondents with some women reporting violent victimization dating back to childhood, by family members, or by partners. Approximately half (*n* = 16) of all respondents had either witnessed violence against women living on the streets or experienced violent victimization as a homeless woman. A woman reported, (*“…There are several women I see with black eyes. At [a women’s shelter], ladies off the street have come in with black eyes and busted lips or something like that…”*) Another woman living on the streets had sought help from extended family members for shelter. In response, she was “hit” and turned away. Women reported a general sense of helplessness and fear for their personal safety while living on the streets. (*“…Because as a woman, I feel like I would not been able to protect myself. I just, I think it is so much worse as a woman…”*)

The female respondent who disclosed her story of sexual assault also noted the normalcy of violence on the streets. (*“…I’ve seen people get jumped, stabbed…it’s bad out here. And, you know people they just watch it happen. People out here when they get too far gone, they get violent…”*) She shared her own history of violent victimization. (*“…I’ve been threatened. I’ve actually been stabbed. You know up in Detroit I got shot. So, I’m like I’m the type of person like I’ve already been through it…”*) She cited her childhood abuse as the reason for her sexual assault experience:
“I got abused at home. Like really bad. So for me it was like, I think, like for me, it was more natural to happen.”

#### 3.1.3. Intimate Partner Violence

The intimate partner violence theme stemmed from interviews with only female participants. Several of the women found themselves displaced overnight by their partners. One female interviewee had returned home from a trip and was locked out by her husband. (*“…I know he had an affair in the past and I’m assuming it’s something like that, so I was basically locked out and I couldn’t get back home. He served me with divorce papers, so that’s how and why I’m homeless…”*) Women escaping intimate partner violence reported continued harassment and threats from their partners who at times utilized custody of their children for leverage. (*“…He’s like, ‘I’m just going to dump you off on the side of the street where you belong.’ And I said, ‘That’s not fair. My son’s here, I just want to see my son.’ He stopped the car; he took my suitcase out of the trunk and drove away…”*) Six of the female interviewees were escaping intimate partner violence with four entrapped in custody battles for their children, requiring legal counsel that our mothers could not afford or struggled to coordinate while securing shelter. (*“..Now he’s claiming that I abandoned my son and…he’s trying to hit me for child support…there’s a custody hearing tomorrow they just told me about today—that I can’t even afford to go to, and he even told me, ‘I’ll screw you over any way necessary,’…”*) Powerlessness was a common thread reported by this group. One woman’s mother had turned over custody rights of her son to her abusive partner, who is not the biological father. (*“…My mom showed up to court and did not even tell me... She signed my son over while I am trying to file a restraining order…I have threatening text messages before all this. This just goes along with why I am homeless.”*) Another woman escaping domestic violence shared her struggles living on the streets:
“For women, it’s not as easy, especially as a victim of domestic violence…. I was out on the street...it was my first time, not finding a place to go… I kept moving because [I] never felt comfortable. I was always afraid to lay down and sleep because I wouldn’t know what would [happen] … I didn’t feel good and [was] always upset and outside. But this time when I left, I would not go back.”

### 3.2. Social and Structural Barriers

Our results found barriers to social and structural systems in the region, including shelter limitations, childcare services, and mental health resources. These subcategories emerged from female participants only. All female participants felt there should be more women’s only shelters. Three women were traveling with children and struggled to access their needs. Without Available and adequate services for women facing homelessness, women expressed being trapped in a cycle that readily permits recurring trauma and keeps them from escaping homelessness. They were forced to live on the streets, making them vulnerable to violent victimization and sexual assault. The lack of access to mental health or other supportive services sometimes interfered with their or the ability of other unstably housed women they knew to process past or current trauma, build trust in social relationships, manage overwhelming stress, and survive on the streets. Eight female participants reported depression, five described hopelessness, nearly all felt despair from isolation, and one experienced past suicide ideation. We posit this contributes to the State of Paralysis. Below, we discuss this pathway by theme: shelter limitations, childcare services, and mental health resources.

#### 3.2.1. Shelter Availability and Limitations

When asked about their barriers to stable housing, the female respondents cited an inadequate number of Available shelters and the high cost of living in the city. (*“…There should be more shelters for women…The majority of these places are just for food, clothing, and shelter. I have seen like two women there. I feel like there should be more housing for them…”*) The *waitlists for the region’s temporary housing and bridge shelters averaged from 8 to 15 years*, though nearly all of these are for men and women. Only one of these shelters specifically serves women and took only 35 occupants at a time in its night dormitory. When asked how many come through the women’s day center, a female respondent stated “100 women or more”, highlighting a critical gap in services because the majority cannot get into the night dormitory.

Women’s only accommodations were said to be important because of the safety issues that women may feel among men, particularly for women escaping intimate partner violence or traveling with children. As one woman said
“…I have gone into shelters for one reason, one reason only, because I have a child. Otherwise I would not have gone into there…I have strayed away from certain shelters because of the unsafety issue, because of certain things that are allowed in shelters that shouldn’t be allowed in shelters…”

Five out of 15 women worked part-time or full-time, yet they struggled to make ends meet. One mother spoke on the challenge of having to work multiple jobs in order to afford rent and the issue of being turned away by landlords. (*“…But it’s just the struggle of having housing that can’t help us, the rent’s too high, we can barely pay it, we can’t just work one simple job, we don’t have enough income, they just won’t accept us, everything is a struggle….”*) She continued:
“The temporary [shelter] wants us to look for apartments when we’re ready. When I look at apartments, it’s like a thousand or something. I’m low income. My son doesn’t make income because he’s a four-year-old. There’s nothing I can do.”

The conditions at the shelters were described as crowded, filthy, and “horrific”. A female responded elaborated, (*“…[Women] over in this center at the night shelter—they don’t shower…I don’t get that. Why would you not want to shower in the morning and night? Why won’t you wash your clothes if you have opportunity to wash your clothes?”*)

Women escaping intimate partner violence face several barriers in accessing shelters. One woman was turned away from a domestic violence shelter because she did not have documentation to prove she was a victim. Another woman lived north of the county and could not find shelters within the vicinity. Below, one of our female respondents spoke on the issue of identifying as both homeless and a victim of domestic violence:
“If you tell them you’re homeless, it’s a mistake I made, because I tried to get into domestic violence over there when he kicked me out, and they said, ‘Well, you’re homeless, so that seems to [be] what your problem is. It’s you’re homeless, so go to a homeless shelter.’…So, they refused to help me…”

#### 3.2.2. Caring for Children

Data on caring for children were derived from both the demographic surveys and the interviews with mothers tackling homelessness. These women faced an additional barrier of ensuring the safety and well-being of their children. This was stressful and often led to difficult choices. One mother could not take time off work to move out of a shelter, so she had to quit her job.
“… I didn’t have a choice. I was supposed to show up to work and move out the same day. I was like I can’t do it, I’m sorry, and they were like we’re going to have to let you go. And I said okay, that’s your final decision, but at the end of the day, I have to take care of my kid so I’m going to get him safe…”

Twelve of the female interviewees were mothers, and three were traveling with their children. One mother and her teenage daughter had been living on the streets for over 7 months. They had reached the maximum of 20 days at one shelter and had to live on the streets as they hung on a waitlist for another.

When it came to welfare programs, one woman described criteria barriers. (*“…And people on WIC, unless you quit your job, you don’t qualify for daycare…”*) Another mother was hesitant to contact social services for welfare benefits, out of fear that Child Protective Services (CPS) would take her son away. (*“…I was scared before. Like people are going to call CPS and take my kid from me…”*) One of our mothers fighting for custody of her son had turned to social services for help and was only given a 1-week voucher for a hotel.

#### 3.2.3. Mental Health Resources

When asked about their current feelings of isolation—*“Can you describe the isolation you may currently feel and how you may want to reduce it?*”—many female respondents became tearful. (*“…I cannot speak right now. I want to cry. I am emotional saying all of this. There is just a lot of judgment out there. I would love to see women get better...”*)

Eight female interviewees stated they were struggling with depression, five described hopelessness, and nearly all felt despair from isolation and loneliness. Suicide was also mentioned. (*“…Well for a lot of us that depression it just gets worse and worse and worse. I’ve gone through a lot of suicide attempts as a result and being out here just kind of drives that home even more….”*)

Five of the women had a history of childhood abuse, with two sharing personal accounts of childhood sexual trauma and how it continues to impact their ability to have healthy relationships. (*“…It certainly interferes with my ability to have intimate relationships with people…”*) They cited wanting to talk to someone regarding their trauma yet struggling to trust service personnel. (*“…I think for most of us, (um), you just really don’t know who to trust, so everybody’s a threat at all times….”*)

Female respondents observed that homeless women often had low self-esteem due to their history of trauma and abusive relationships. (*“…There are a lot of women that are embarrassed. They look like they have been abused and talked down to… the majority of them are just depressed. Their self-esteem is so low because of men or other people…”*) For the mothers in our study, the stress of securing custody of their children was a prominent mental health theme. (*“…I am trying to keep my heart rate down because of all this stress. I do not want to ever lose [my son]…”*) Women escaping intimate partner violence described overwhelming seclusion. (*“…I don’t have any family... I don’t really have a support system. So, that’s how I ended up in this situation…”*) Similarly, the woman who had been sexually assaulted felt she had no one to talk to about the incident, which she cited as the “scariest experience” in her life. (*“…I didn’t talk to nobody, I didn’t have any friends…”*) One mother was 18 years old when she became pregnant and homeless. She described her mental state as “scared out of [her] mind.” (*“…I felt alone. I felt like I had no one in this world. Until I looked down and I just can’t give up. I have a little boy coming on the way…”*)

Several women felt that the longer people lived on the streets, the more socially and emotionally despondent they became. (*“…I couldn’t regulate my emotions very well…”*) The woman who had been sexually assaulted observed that chronic homelessness resulted in severe mental health symptoms. (*“…The depression and all that like it gets worse the longer you are out here. And then you get the people that are screaming at the top of their lungs at the air or screaming at their reflection or losing their mind…”*) She emphasizes her own sense of hopelessness:
“I guess the homeless mental illness is all wrapped together or whatever but, but uh, …cause like really, like at the end of my life on the street, like, like I was pretty despondent. I had given up on everything.”

Many of the women in our study recommended more mental health services and counselors. Some felt discriminated by services and resources, which fueled their distrust, hopelessness, and fears. (*“…Sometimes you just need someone to talk to or you need somebody to understand…A lot of the times they kind of discriminate against us and refuse to help us…”)* Another respondent spoke on the fear of being rejected or turned away. (*“…[People who are homeless] don’t want to be judged, they don’t know if people are actually willing to help them. I’ve been through that. It’s scary…”*) While some shelters provided counselors, respondents felt that the counselors did not provide the quality of care that could be helpful to those experiencing hardship. They emphasized the need for “compassion training,” particularly as they have faced so much trauma. (*“…I think that they should have some compassion training…’cause I feel like they don’t really understand our situation and I really don’t feel like they particularly like us…”*) A mother shared one of her most distressing experiences, and how a counselor could have helped:
“There are counselors in the day shelter, there are counselors in the night shelter…It would, ideally, it is supposed to work that you are, that when you come in, you are given a counselor in the day shelter and one in the night shelter as well…But in reality, it’s not. Like I was never spoken to in the night shelter. It would have been nice. I mean there was one time in particular I came in from court and it had not gone well for me and I was just devastated and I didn’t necessarily feel like reaching out so it would have been nice if the person assigned to me would have said “oh wow, it’s your court date, do you want to talk about it?”

Mental health and other social services were found to be critical gaps to serving this vulnerable population. Without access to adequate social services, women facing homelessness felt ensnared in their state of crisis.

#### 3.2.4. A Cycle of Perpetual Vulnerability Theory

Emerging from these themes, we developed a *Cycle of Perpetual Vulnerability Theory* (Figure 1) for women feeling trapped by homelessness. The themes suggest that women facing homelessness are often disproportionately preyed upon for violent acts and/or sexual exploitation. Half of male and female participants witnessed violent acts against women or experienced unwanted sexual harassment and suspected human trafficking. When women are preyed upon, it can form a pathway of a *target phenotype*, where normalized violence and history of trauma from being abused can make them more susceptible targets of abuse on the street. Their violence victimization combined with history of intimate partner violence and childhood maltreatment is defined in our pathway as *iterated trauma*. Both pathways relate to a *state of paralysis*, where women experiencing homelessness feel paralyzed, unable to escape vulnerability due to social and structural barriers, such as lack of adequate shelters and mental health resources. They find it challenging to secure safe environments, to form supportive social relationships, and to acquire services that could lessen their vulnerabilities. Figure 1 displays a *Cycle of Perpetual Vulnerability* with three pathways derived from our findings and knowledge of the gaps in the literature.

Thus, our analysis generated theoretical principles underlying homelessness for women, including a cyclical interdependency of relational pathways, influenced by violence against women, social and structural barriers, and exacerbated mental health issues.

## 4. Discussion

This study fills a theoretical gap by constructing a *Cycle of Perpetual Vulnerability*, which proposes three relational pathways: *target phenotype, iterated trauma, and state of paralysis* that leaves women in a disempowered state, stemming from mental, social, and structural factors. Although the study participants were diverse, the vulnerabilities of women facing homelessness resembled a *target phenotype* through being perpetrated against or victimized. Such chronic abuse combined with the relationship between mental health issues and homelessness led to *iterated trauma* for many, often leaving women in a *state of paralysis*. Paralysis is the psychological impairment and the inability to make or achieve healthy and safe choices for one’s well-being caused by mental health issues and social and structural barriers to services.

### 4.1. Violence against Women—Target Phenotype

Violence among people who are homeless is pervasive—ranging between 14% and 21%, though certain characteristics can increase one’s risk for violent victimization [20]. Studies find that the most significant risk factor for adult violent victimization is a pattern of childhood abuse and maltreatment [21]. According to a study examining violent victimization among 800 homeless women and a comparison sample of 100 homeless men, over half of the women reported minor childhood maltreatment and nearly half reported severe maltreatment; 30% had cited violence as their reason for homelessness [21]. Women in the study reported higher cases of rape—over half, compared to 14% of men who reported rape [21]. Women also reported higher for certain types of physical assault, including, pulled hair, slapped, hit, and choked [21]. While homeless men and women are both at risk of physical violence and assault, studies have shown women to be at disproportionately higher risk for sexual assault. Compared to women of the general public, women who live on the streets are more at risk for vulnerabilities [21].

Experts cite that perpetrators may target homeless women for various reasons—their exposure and visibility on the streets, knowing where they reside or when they are alone, and their comparably smaller size and inability to protect themselves [21]. In another study sampling 974 homeless women in Los Angeles, one third of the women experienced major violence, including being kicked, bitten, hit with a fist or object, beaten up, choked, burned, or threatened or harmed by a weapon [22]. Researchers found that women reporting childhood maltreatment correlated with longer episodes of homelessness and major acts of violence [22], suggesting that violence and homelessness inexorably become routine. That is, homelessness accompanies instability, triggers, and an absence of safety, which may look and feel familiar to women having a history of childhood maltreatment.

Recent findings indicate that women continue to be at greater risk for sexual victimization compared to men, with sexual abuse as a child being positively associated with revictimization and negative mental health outcomes [23]. As one female respondent in our study reported, being sexually assaulted was “natural to happen” due to her own history of childhood abuse. However, further investigation on the risks of sexual assault for men who are homeless are perhaps needed, as well as more studies on human trafficking for both men and women, as both of these may be under-reported.

Existing literature confirms the normalization of sexual assault for women who have a childhood history of violence. In a qualitative study of violence against eight American Indian and Alaska Native women, the women self-report that incidences of intimate partner violence and sexual assault are a matter of “when” they will occur and not “if”, having normalized these events from witnessing it or experiencing it throughout their childhood [24]. The study corresponds with this study’s theory that pathways are interdependent. It argued that it is not one sole factor, but rather the combination of factors that contribute to an increased risk for sexual assault. As with our pathways, factors for sexual assault against American Indian and Alaska Native women include a history of trauma, lack of proper resources, and normalization of violence [24]. This normalization and history of violence may contribute to the target phenotype described in other women facing homelessness.

### 4.2. Mental Health—Iterated Trauma

As with our study, previous literature has noted the interdependent role that mental health symptoms and disorders plays with homelessness. In a qualitative study with in-depth interviews among 13 homeless women struggling with mental illness, researchers found trust and trauma to be an interrelated theme [25]. Similarly, several women in our study had reported unresolved childhood maltreatment that seeped into their ability to maintain healthy relationships and develop trust. In fact, these mental health symptoms could be under-reported due the stigma of mental health, and perhaps limited ability to self-assess for symptoms. Another study found comorbidity between psychiatric conditions and violence among unstably housed women, reporting that those with dual diagnoses experienced higher risk for violent acts [26]. In another study sampling 703 homeless women, those who experienced violent victimization struggled with severe negative mental health outcomes, and often reported poorer quality of life compared to their male counterparts [27]. The study corresponds with our findings where women who experienced sexual assault and had a history of childhood abuse had also reported suicide ideation and negative mental health outcomes. In general, all patrons in our study felt their city needed more mental health services for those who were homeless, concluding that existing services were insufficient; they were often discriminated against and judged for seeking services.

In our theory of the Cycle of Perpetual Vulnerability, we suggest that women who are preyed upon for violence victimization or sexual exploitation often suffer from recurrent mental health symptoms, including post-traumatic stress disorder, depression, anxiety, suicide ideation, among others. These symptoms are further augmented by any history of childhood maltreatment and intimate partner violence. In a study of homeless women veterans, similar pathways to homelessness were found among the veterans, including trauma or abuse events, childhood adversity, and mental health problems [28]. Like our study, contextual factors to homelessness include barriers to care [28], supporting our study’s aim that social and structural barriers can disable women from escaping homelessness. These findings coincide with our pathway, “state of paralysis” that prevents vulnerable women from having access to services, resources, and care that can help with overcoming homelessness. Thus, the pathways of “iterated trauma” and “state of paralysis” become cyclical with target phenotype.

### 4.3. Social and Structural Barriers—State of Paralysis

Our sample of people struggling with stable housing unanimously cited that the city had a shortage of shelters. Mothers are particularly vulnerable because they have the high stakes of caring for the well-being and safety of both themselves and their children. Trauma from violence and harassment, unmet medical needs, and residency instability are prevalent issues among homeless mothers traveling with children [29,30]. They often lack the infrastructure to support their economic and social needs [31]. Many single mothers often forced to make difficult choices—such as paying for childcare or going to work, and this is further complicated for mothers experiencing homelessness. In lieu of state provided childcare, shelters, or more options for low-income housing for mothers facing homelessness, a social network such as family or friends would be helpful. However, homeless mothers tend to have smaller support networks compared to low-income mothers and women of the general public [31]. Most of our mothers were turned away by family and friends during their time of hardship, an experience that also wounded them emotionally. Historical tensions between families and child welfare systems also interfere with this population’s ability or willingness to seek services [32]. In our study, one mother had avoided welfare programs because she was fearful that Child Protective Services would take away her son due to being unstably housed. Similarly, the women in our study that were fleeing intimate partner violence reported fear and intimidation of their partners, as well as powerlessness in maintaining custody of their children. They were often in a state of paralysis due to financial insecurity, absence of social capital, and emotional impairment. These stressors exacerbate the homeless experience for women. Mental health and other social services are critical to serving this vulnerable population. Without access to social services, women facing homelessness can feel ensnared in their state of crisis.

Our Cycle of Perpetual Vulnerability theory views this absent infrastructure as a major contributor to *a state of paralysis.* Appropriate shelters and services can serve women two-fold: (1) women and their children are no longer on the streets, therefore less likely to be a visible and a target for violence, to be exposed to violence, and to be in an environment where they are approached for sex trafficking or exploitation; (2) agencies with adequate staffing could protect women by patrolling shelters and regulate socially sanctioned acts, interfere with inappropriate behavior, and serve as a point-of-contact for protection should women and their children feel at risk. In addition to shelters, resources such as childcare, job training programs, or low-income housing that prioritizes homeless families would be instrumental in serving mothers facing homelessness, particularly if they have been turned away, exhausted resources, or lack social capital for support or opportunities for economic mobility. Shelters, state relief for mothers such as childcare services, job training, low income housing, and standardized policies for domestic violence eligibility could all help protect mothers and their children from a state of paralysis.

## 5. Limitations

This study was limited by its sample size and convenience sampling, but intended to expand on the existing literature on women facing homelessness by contextualizing the results into a new framework reflective of the issues women seem to perpetually face. It was also difficult to find women who directly experienced human trafficking; more research is needed on this hidden and vulnerable population. Future research could also further explore the protocols in shelters that pose social and structural barriers. For example, our study found that one woman seeking domestic violence shelters was turned down due to lack of proof of domestic violence. Future research could also investigate the efficacy of the cycle of perpetual vulnerability with larger populations. The study was conducted in an urban southern California region, thus may reflect some characteristics unique to the region. Some participants helped recruit others from a specific women’s shelter, so that more participants came from one shelter, though they all frequented the library and had varied experiences being homeless prior to living in the shelter.

## 6. Conclusions

Between 2009 and 2019, the United States experienced an overall 10% decline in homelessness, though the figures have been steadily rising since 2016 [33]. In the last year, homelessness increased by 2.7% [34]. California experienced the largest growth of homelessness and has the largest concentration of individuals experiencing homelessness; 71% of its homeless population is unsheltered [1]. Among top five urban areas for homelessness, New York City ranks highest, but three of the four remaining cities are in California. These numbers highlight the pervasiveness of homelessness and the critical response required to combat the crisis not only in the US, but globally, where over 1.6 billion lack adequate housing [35]. Among those who are struggling with homelessness, changing demographics include race/ethnicity, sexual orientation, and gender [1]. The US has experienced a rise in the number of homeless women over recent years, representing 39% of the population in 2019 [1]. Women who are facing homelessness are at a disproportionately greater risk for violent victimization, sexual exploitation, human trafficking, among other acts of trauma.

The objective of this study was to explore and investigate the experiences, perceptions, and critical needs of women experiencing homelessness. Our aims included, (1) investigate vulnerabilities for women, including violence, sexual assault, and sex trafficking risks; (2) identify existing barriers for women, including shelters, childcare, and domestic violence resources; (3) explore the relationship between mental health and homelessness for women. We found that the inadequate number of shelters and supportive services increased the risk of violence, sexual assault, and mental health trauma for women facing homelessness in this large U.S. city. We further proposed a theory known as the Cycle of Perpetual Vulnerability, which informed the interdependent variables known as “Target Phenotype”, “Iterated Trauma”, and “State of Paralysis” in the experience of homelessness for women. The theory suggests that women who experience repeated acts of violence or trauma are targeted and then re-traumatized, a cyclical pattern that makes it difficult for women to escape homelessness.

Furthermore, by conducting qualitative interviews, we were able to hear the personal stories of women experiencing homelessness and to give voice to this hidden population. In the same vein that our women shared accounts of unimaginable hardship and trauma, they also shared their resilience and their strengths.

Based on our findings, our recommendations are to increase the number of women-only shelters, supportive services, and research on women experiencing homelessness. It was unanimously cited among the female participants that there is an unmet need for women-only shelters, which corresponds with the rising statistic of unsheltered women and girls in the U.S. Many of the women in our study experienced intimate partner violence, childhood maltreatment, and violent victimization, thus women-only accommodations can help address safety concerns in co-ed shelters. Supportive services such as childcare services, job training, low income housing, and access to mental health counseling would serve mothers and women experiencing homelessness. Many of the women in our study were mothers, some traveling with children or fighting for custody of their children, and many cited the need for more assistance, such as subsidized or state provided childcare and mental health services. Lastly, more research is needed on this growing subpopulation within the homeless community. As mentioned above, qualitative research provides a space to explore and learn from the direct accounts and perceptions of those being served [36,37,38]. Women have historically experienced socio-cultural marginalization. The additive of homelessness exacerbates the powerlessness they may feel. Further research capturing their voice and needs can empower women, while also addressing gaps in services.

As homelessness continues to grow, a response needs to evaluate the issue as a whole—what experts deem as a “whole person—whole agency—whole community—whole system approach” using both micro and macro interventions in leadership, policy, and integrated services [39]. Homelessness is a global crisis that has persisted in both developed and impoverished nations alike, highlighting the need for a multifaceted approach among social workers, policymakers, clinicians, organizers, and communities.

## Figures and Tables

**Figure 1 ijerph-17-05985-f001:**
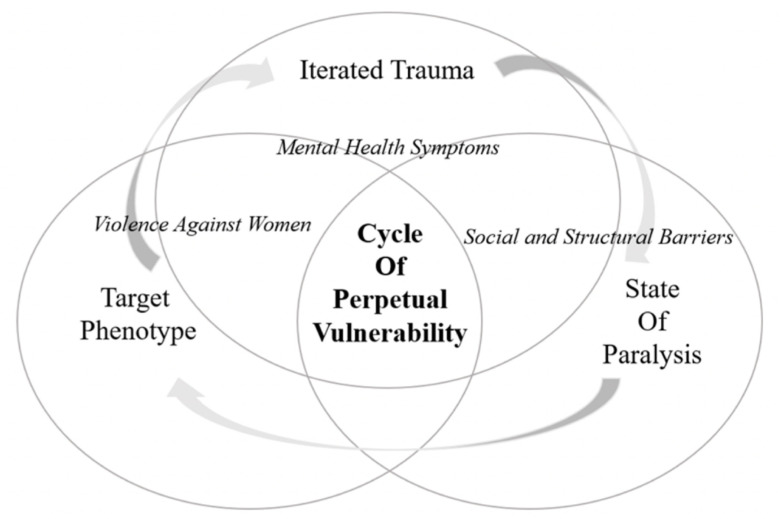
A theory of a Cycle of Perpetual Vulnerability.

**Table 1 ijerph-17-05985-t001:** Socio-demographic characteristics, unstably housed study participants qualitatively interviewed at a large urban library in southern California (*n* = 32), derived from socio-demographic surveys.

Characteristics *n* (%)	Males (*n* = 15) (%)	Female (*n* = 17) (%)	Total (*n* = 32) (%)
**Race/Ethnicity**			
White	6 (40)	6 (35)	12 (38)
Hispanic/Latino	1 (6)	6 (35)	7 (22)
African American/Black	4 (27)	3 (18)	7 (22)
Other/Mixed Race	3 (20)	3 (18)	6 (19)
**Sexual Orientation**			
Heterosexual	14 (93)	14 (82)	28 (88)
Lesbian/Gay/Bisexual	1 (7)	3 (18)	4 (13)
**Disabled** (physical, mental, or educational)	8 (53)	10 (59)	18 (56)
**Housing Status**			
Living on the street/Homeless	9 (60)	7 (41)	28 (88)
Shelter	2 (13)	8 (47)	11 (34)
Staying with a friend or couch surfing	2 (13)	1 (6)	3 (9)
Single rent	2 (13)	2 (12)	3 (9)
**Seeking domestic violence shelter ***	0	6 (35)	6 (19)
**Has dependents/children**	5 (33)	13 (76)	13 (41)
**Has children living with them ***	0	3 (18)	3 (9)

* Collected from individual in-depth interviews.

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
