# Peer review of "Cycle of Perpetual Vulnerability for Women Facing Homelessness near an Urban Library in a Major U.S. Metropolitan Area"

_ijerph, 2020, doi:10.3390/ijerph17165985_

Round 1

Reviewer 1 Report

The main point of this paper seemed to be to provide support for a theoretical framework to understand homeless women's victimization and vulnerability.  Unfortunately, the methodology didn't seem to be clearly described or linked to the framework and left a number of questions as to what was actually in the interview guide, and in the survey, and how the qualitative analysis was conducted. Reporting of results was also confusing.  It was not reported clearly how many women resided in a shelter or on the street, how many were accompanied by children or not, how many were domestic violence victims etc.  A table giving basic characteristics of the sample would have been helpful.  Further, when results were reported, it appears that views of the male sample were included but its not clear sometimes when it is a male observation or a combination of  views of participants across genders that the authors are summarizing.    

Some more detailed points: Methods: On the qualitative framework being used- its not clear if the authors were using a thematic analysis, a phenomenological approach, grounded theory? On the interview guide and survey-how many total questions, what were the categories, what were the follow up questions?  The brief explanations seems to overlap somewhat on what was covered. Were questions grouped to investigate the proposed framework or did the framework arise out of the data? On analysis of transcripts: were coders blind to the study framework if it was developed before the study? Was a qualitative software program used?  How was interrater reliability carried out? 

Results:  the model is reiterated here instead of just presenting the data; as noted earlier a table of the main characteristics of the participants would be helpful.  Also, which data come from the survey and which from interviews. Reporting in each section should clarify which themes or ideas were generated by the whole sample and which by women only. The titles of the sections are confusing since they state two different possible themes like shelter limitations vs barriers to housing (assuming that means permanent housing); caring for children vs. making money. 

Discussion: In the discussion, the first mention is made of interviews in front of children. Earlier in methods it is not described how many women this affected.  It should be discussed in methods and in limitations but not in this section.  Social and structural barriers are not defined.  Comments on line 510-11 in the discussion about 'state relief, policies..." didnt come up in  the results reports directly although could be inferred, but its not clear what types of relief or policies are being recommended. Do the authors mean shelter policies like limits on stays? Requirements about access to domestic violence shelters?

Finally, there are a number of awkward and unclear sentences throughout.  A few are: line 82:  "...women who have been perpetrated over time..."; line 99: "notional principles"; line 108: "public libraries have become an epic center..."; line 445: ..."violence increased with each additional diagnosis..." 

Reviewer 2 Report

1) the abstract needs to be rewritten, given that it contains too much information, which should be in the 'methodology or methods' sections. The objective of the abstract is to inform, the aim, relevance, methodology (briefly, rather than too much details, e.g. this study takes a qualitative approach or 'this study is based on interviews...), findings and the social relevance, in that order and in a more powerful way to capture the readers interest...

2) there are too many key words: better to aim for max of 6.

3) The legal definition/classification of the term 'homeless' needs to be further explored or at least clearer explained in a footnote, given that the lack of global definition impacts on the 'identification' of homeless.

4) The same (1) in relation the term 'vulnerability' (particularly because you use this term in the title), which in some contexts (e.g. the UK) depends on the local governments judgment and, therefore, it can be used (subjectively) to identify a person who can become or is homeless.

5) Why figure 1 is part of the introduction? Is this part of the literature review or the methodology?

6) You need to further 'explain' how the data was analysed (methodology) in the section 'data analysis'. you can display the Qs in an appendix, rather than here or as part of the sub-heading. The purpose of subheading is to be short and interesting to guide the reader smoothly, rather than leaving her/him guessing or to provide too much info as you did here).

7) who performed the interviews (are the researcher or interviewers as you indicate the authors?) 

8) I find the 'limitation' section rather long (which is certainly the case, but it nearly jeopardises your work). Instead, you could Use the number of words to provide clearer explanation on the phenomenon within the context of your study.

9) The conclusion (which I would suggest to name 'final considerations' terribly short. Instead, you need to remind the reader of the aim and relevance of your study, link it to the literature review, set the contributions very clearly and further research too. Then link it to the societal/international issue, since homelessness is not only perpetuating but also increasing internationally....try the to address the 'so what?' question when you end this section, so it provokes the reader engagement.

10) Good luck in improving your paper. This is an interesting subject that needs to be further explored.

Round 2

Reviewer 1 Report

Authors have made substantial changes to the manuscript that have considerably improved communication of the study.  However there remain a number of issues in the communication of the study methods and findings.  It also appears that the way questions were asked and/or the selection of the results to report end up in a bit of overinterpretation of the findings of the study. 

First, it still remains unclear how the male respondent responses were used in some of the findings.  For example, Section 3.2-do all these findings relate only to women's responses? While for most of the sections there is a referral to female only respondents, but under shelter limitations, there also is a reference to both male and female respondents talking about the poor sanitation in facilities.  Also on line 256, female respondents are described as 'female contenders', not interviewees or respondents which is an odd phrase. 

Throughout the findings and discussion there is referral to 'mental health' as causing challenges, rather than mental health issues, difficulties, or problems causing challenges. (Ex lines 396, 449).

In the discussion (line 372) there is a conclusion that women suffer 'disproportionately" from victimization but men are also mentioned.  Overall the authors have done better at explaining why men were reported to be part of the sample, but many places its still unclear what their contribution is and there is no evidence cited as to why men didnt also suffer from similar victimization. I would suspect that men experiencing homelessness experience very similar histories of family abuse, and being victims of violence on the streets, including sexual violence, and I am sure there are references documenting this. So it just needs to be more clearly delineated that the men were interviewed about their views of homeless women's victimization, or if they told their own stories and if so, more clarity about the different themes arising from men and women needs to be drawn.  It may be the authors are aiming to write a different manuscript about men in the sample, but if so, perhaps its simpler to just include women's views in this paper?

Line 379: "challenging to assess ..for safety" "challenging to seek services"- I derived the sense from the quotes that women had a good sense of their environment not being safe, and that they were seeking services.  The problem is that the existing services are not enough or not well matched to their needs, and thus they cant get out of the cycle of homelessness which brings iterations of new trauma-which promotes the feelings of isolation and helplessness.  I dont think this section interprets well the data that have been presented and should be rethought since its central  to the model the authors are trying to promote.

Some other smaller issues:

The abstract summary of the gender identity of the sample is not consistent with the Table now provided.

The insertion of Section 8 in the introduction defining homelessness as not permanent housing is inaccurate.  Section 8 is a permanent housing program.

Line 348 describes discrimination 'from' services instead of discrimination 'by' services. 

Line 366-367 should be moved to the discussion, however it refers to both mental health and social services while the results seem to focus mainly on mental health issues.

Throughout the findings and discussion there is referral to 'mental health' as causing challenges, rather than mental health issues, difficulties, or problems causing challenges.

Line 378 refers to forming "important social relationships." Dont you mean "safe"  or 'supportive" social relationships instead of negative or dismissive ones?

Line 442: should be 'discriminated against'

Line 463 refers to 'work place policies for mothers facing homelessness' and this is reiterated later.  Not sure what this means. Paid leave to take care of finding an apartment specifically? Paid personal time that could be used at employees discretion? In general low wage workers do not receive paid leave for anything, including illness, so this is a larger policy issue that could be pointed to.

Finally, in the discussion are citations about growth of homelessness worldwide.  This is unnecessary and misleading.  First, there are over 45 million world wide refugees.  So why is the citation only 1.5 million homeless? Also Iceland is hardly a good example of one country where homelessness might be growing since the whole population is only about 364,000-smaller than the 1.4 million in San Diego- so percentage changes in homelessness there would be totally noncomparable. Sticking with the local area or other major US cities would be fine. 
